

# Analysis of the buildup of spatiotemporal correlations and their bounds outside of the light cone

**Nils O. Abeling**[⋆]**, Lorenzo Cevolani and Stefan Kehrein**

Institut für Theoretische Physik, Georg-August-Universität Göttingen,
Friedrich-Hund-Platz 1, 37077 Göttingen, Germany.

⋆ nils.abeling@theorie.physik.uni-goettingen.de

## Abstract

In non-relativistic quantum theories the Lieb-Robinson bound defines an effective light cone with exponentially small tails outside of it. In this work we use it to derive a bound for the correlation function of two local disjoint observables at different times if the initial state has a power-law decay. We show that the exponent of the power-law of the bound is identical to the initial (equilibrium) decay. We explicitly verify this result by studying the full dynamics of the susceptibilities and correlations in the exactly solvable Luttinger model after a sudden quench from the non-interacting to the interacting model.

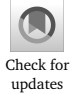
---

## 1  Introduction

The rising interest in the out-of-equilibrium dynamics of physical systems has posed the question of the propagation of information in many-body quantum systems. While relativistic models intrinsically contain the notion of a maximal speed, i.e. the speed of light, it was historically unclear whether there is an equivalent bound in generic non-relativistic models. A milestone in this context was achieved by Lieb and Robinson in 1972 [1]. They demonstrated that the dynamics of a lattice system with finite-range interactions is *effectively* constrained inside a region of the spatiotemporal plane, with exponentially small "leaks" outside of it. As a consequence, any perturbation effect requires a finite amount of time to spread from one point of the system to another. Mathematically, the theorem provides a bound for the operator norm of the commutator of two local observables $O_A$ and $O_B$ defined over two disjoints sets $A$ and $B$ and reads

$$||[O_A(t), O_B(0)]|| \leq c N_{\min} ||O_A|| ||O_B|| \exp\left(-\frac{L - v|t|}{\xi}\right), \tag{1}$$

where $O_A(t)$ ($O_B(t)$) denotes the time-evolved observable with norm $||O_A||$ ($||O_B||$), $N_{\min} = \min(|A|, |B|)$ and $c$ and $\xi$ are additional constants that depend on the interaction, the local Hilbert space dimension and the lattice structure. $L$ is the distance between the sets $A$ and $B$ (smallest number of vertices connecting the two regions) defined over the lattice. The parameter $v$, which has the dimension of a velocity, is called Lieb-Robinson velocity. The value of this important parameter is not fixed by the theorem, but depends on the lattice structure and the interaction [2]. The previous expression which is also called "Lieb-Robinson bound" (LR bound) tells us that the commutator between the two operators is exponentially suppressed for $L > v|t|$. Keeping the distance $L$ fixed, the value of the bound grows in time until $t = L/v$ where it becomes $O(1)$. It thereby divides the full spatiotemporal plane into a part *inside* and *outside* of a light cone which is defined by the Lieb-Robinson velocity $v$. Since the "leaks" outside of the light cone practically vanish, the two observables *effectively* commute in this region (also called "space-like" region in analogy to relativistic theories). Therefore, the finite velocity $v$ defines the fastest possible information propagation.

Historically, the theorem first only applied to translationally-invariant quantum spin lattices with short-range interactions until some of the requirements were loosened by several improvements and generalizations [3–6]. In its most recent version, it does not require a finite local Hilbert space, since it was extended to systems with an infinite local Hilbert space, for example lattice bosons [4–8]. Moreover, it can be generalized to long-range interacting systems [2,9–12], where a faster-than-ballistic propagation, with an algebraic or even logarithmic light cone, is allowed. However, the propagation observed in some specific models appears to be significantly slower than the one allowed by the bounds [13–16]. Despite the interest for the latter, we will focus in this paper on short-range interacting systems, where a ballistic light cone is present and postpone generalizations with lesser requirements to further studies.

In the discussion about the LR-bound it is important to notice that the commutator of two observables as it appears in Eq. (1) is closely related to linear response theory (see for example Ref. [17]). It measures the response of a system to a local perturbation: the system is shaken at one spatial point and the corresponding effect is observed at some other point later in time. Therefore, the commutator is also known as the susceptibility. In a simplistic semi-classical argument one models the spreading of the signals to be governed by quasiparticles that move through the system [18]. The light cone velocity, which was a non-fixed parameter in the LR result, can then be determined by the microscopic theory to be equal to the maximal group velocity of the excitations defined in the system. It must be noted that this concept only holds in very basic models.

Experiments, however, do not only focus on susceptibilities, but they can also access other quantities like correlation functions of the form $|\langle O_A(t)O_B(t_0)\rangle|$. They are directly connected to the anticommutator $\{O_A(t), O_B(t_0)\}$ (see section 6). Notable developments which focus on the study of the dynamics of the correlation functions have been achieved in cold-atomic gas experiments [19, 20]. Additionally, the spreading of correlations has been studied using several advanced numerical techniques like tDMRG [21–25], time-dependent variational Monte-Carlo [13, 26], and artificial neural network techniques [27]. Finally, exact analytic results on the dynamics in solvable models [28–35] have helped to understand the mechanisms and to build up a physical picture where one can interpret correctly the observed phenomena. The original LR-bound as in Eq. (1), however, does not provide information about the behavior of such quantities which strongly depend on the initial state. Yet, few mathematical bounds for the correlation function which rely on assumptions about the initial state have been found. They require either an exponential [8, 36, 37] or algebraic [38] decay of initial correlations or a spectral gap above the ground state. All these results are derived using the LR-bound in different ways.

The main difference between the commutator and the anti-commutator, i.e. the susceptibility and the correlation function, is the behavior outside the light cone. The correlation function in fact exhibits "leaks" in that region, meaning that correlations are non-zero from the very beginning of the time evolution. This does not contradict the main result of the LR bound, becaus ecorrelations between two space-like separated regions do not imply superluminal propagation of information, like in the EPR-paradoxon [39]. This difference between commutator and anticommutator has been shown explicitly in the Kondo model [17]. There, one can see a light cone in the susceptibility (with exponentially small "leaks"), while the correlation function has algebraic "leaks" outside of it. The extension of the LR bound to non-equal-time correlations and its relation to the initial state are the main results we want to present.

This paper is organised as follows: In the next section we derive a bound for the non-equal-time correlation function in the case where the initial correlation function is algebraically decaying. We will demonstrate that also for this case the correlations have a light cone with a velocity determined by the Lieb-Robinson velocity. This generalizes other results present in the literature for equal-time correlations [36, 38]. Moreover, we connect the leaks outside of the light cone with the initial (equilibrium) correlation function.

Thereafter, we derive exact expressions for the dynamics of a system described by a Luttinger liquid. We calculate and discuss the correlation buildup and decay regimes for the commutator and anticommutator of the density-density correlation for both zero temperature and a thermal initial state.

## 2 Bound for the correlation decay outside the light cone

In this part we prove that the correlation decay outside of the light cone is defined by the spatial decay of the initial (equilibrium) correlation function. Bounds for the correlation functions at equal time have already been found for both exponentially [36] and algebraically [38] decaying correlations in the initial state.

We extend these results to the non-equal time case, i.e. for $|\langle O_A(t) O_B(t_0) \rangle_c|$. The $c$ subscript indicates that we consider the connected correlation function defined as $\langle O_A O_B \rangle_c := \langle O_A O_B \rangle - \langle O_A \rangle \langle O_B \rangle$. Moreover, we show how the space-like region decay is closely connected to the initial correlations. Physically, the theorem states that any leaks outside the outer light cone defined by the LR velocity are bounded by the initial (equilibrium) correlations which have been "dragged" during the time evolution, see Fig. 1.

The following results hold for all models for which the famous Lieb-Robinson (LR) theorem in its form displayed in Eq. (1) applies. In the derivation we consider a quantum spin lattice with short-range interaction (also called "local Hamiltonian"). As previously mentioned, it is possible to loosen this condition to a more generic setting which anyway does not change the main message of our result [4–6]. One example is to consider a continuous system (not on a lattice). In section 5 we will explicitly demonstrate how the bound naturally arises in the case of the continuous Luttinger model. For the moment, we postpone the generalization to systems where more general LR-bounds have been found (for example for a system with long-range interaction) to further studies.

In order to bound the correlation function we assume that the time evolution is governed by a local Hamiltonian and that the initial state has the property that all connected correlation functions decay with a power law (algebraically) given by

$$|\langle O_A(0) O_B(0) \rangle_c| \leq \mathscr{C}^{\mathrm{ini}}(L) := \frac{\tilde{c}}{\tilde{a}^\beta + L^\beta}. \tag{2}$$

Here, $O_A$ and $O_B$ are two normalized ($\|O_A\|, \|O_B\| \leq 1$) operators which initially act on the compact disjoint supports $A$ and $B$, which are apart by the distance $L$. The additional parameter $\tilde{a}$ defines a positive cut-off if $L = 0$. Following Ref. [36], we now study how fast the effective part of the time-evolved operator $O_A(t)$ ($O_B(t)$) grows in space. A different way to bound the equal time correlation function using the LR bound can be found in Ref. [38]. Mathematically, it is clear that $O_A(t)$ requires the full support over the entire system, because the unitary evolution contains all powers of the Hamiltonian. Yet, we will exploit the LR-bound that tells us that *effectively* it only acts on a finite region which grows linearly with time. Therefore, we define an operator, $O_A^{l_A}(t)$, that acts on the region $A$ extended by $l_A$ sites in all directions like $O_A(t)$ and outside of this region as a trivial unity. The outside region of the ball is denoted by $S$. It is defined by

$$O_A^{l_A}(t) = (\mathrm{Tr}_S(\mathbb{1}_S))^{-1} \, \mathrm{Tr}_S(O_A(t)) \otimes \mathbb{1}_S. \tag{3}$$

The normalization makes sure that the norm is given by the inside part. Additionally, $\|O_A^{l_A}(t)\|, \|O_B^{l_B}(t)\| \leq 1$ for all $t$. One can now use the Lieb-Robinson bound to prove that the operator norm of the difference of the full observable and the one only acting inside the ball is bounded by

$$\|O_A(t) - O_A^{l_A}(t)\| \leq c|A| \exp\left(-\frac{l_A - \nu|t|}{\xi}\right). \tag{4}$$

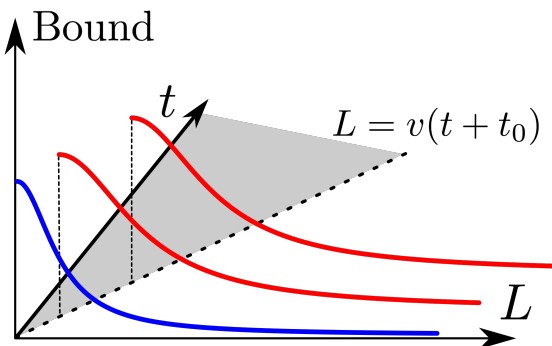

Figure 1: This sketch shows in red the bound (right-hand side of Eq. (8)) over the spatiotemporal plane outside of the light cone. One notices that its scale is different from the initial correlation depicted in blue (Eq. (2)), however, the decay exponent is identical. The inside of the light cone is marked by the grey area and is basically unbounded.

One notices that the difference is basically unbounded (exponentially large) if $l_A < vt$, i.e. inside the light cone and exponentially decreasing, if it is outside. Hence, this states that *effectively* $O_A(t)$ only acts in the region $A$ enlarged by $l_A$ in all directions (if $l_A$ is chosen appropriately). With this result one finds an upper limit for the connected correlation function at different times:

$$|\langle O_A(t) O_B(t_0)\rangle_c| \leq 2c|A| \exp\left(-\frac{l_A - vt}{\xi}\right) + 2c|B| \exp\left(-\frac{l_B - vt_0}{\xi}\right) + |\langle O_A^{l_A}(t) O_B^{l_B}(t_0)\rangle_c|. \quad (5)$$

It is important to note that we have two light cones with an identical velocity $v$, but different final times $t$ and $t_0$. In the next step the last term which by construction only acts inside the two compact balls is bounded using the assumption Eq. (2). The initial support of $A$ has grown by $l_A$ after time $t$ and of $B$ by $l_B$ after time $t_0$ in all directions. As a consequence the distance between the two regions shrinks by $l_A + l_B$ such that one assumes that for all times

$$|\langle O_A^{l_A}(t) O_B^{l_B}(t_0)\rangle_c| \leq \frac{\tilde{c}}{\tilde{a}^\beta + (L - (l_A + l_B))^\beta}. \quad (6)$$

Plugging this into Eq. (5) one gets

$$|\langle O_A(t) O_B(t_0)\rangle_c| \leq 2c\left[|A| \exp\left(-\frac{l_A - vt}{\xi}\right) + |B| \exp\left(-\frac{l_B - vt_0}{\xi}\right)\right] + \frac{\tilde{c}}{\tilde{a}^\beta + (L - (l_A + l_B))^\beta}. \quad (7)$$

For $t = t_0$, this result is the same as the one found by Kastner in Ref. [38] for algebraically decaying initial states up to different constants. It is important to note that $l_A$ and $l_B$ need not be constant, but must fulfill $l_A + l_B < L$. In this way the supports of $O_A^{l_A}$ and $O_B^{l_B}$ do not overlap. We then hit the light cone if $l_A + l_B = L$. Generally, it is not possible to minimize the bound by optimizing the parameters $l_A$ and $l_B$ in Eq. (7). However, an approximated solution which works for many models and cases with short-range interactions can be found. It relies on certain assumptions about the behavior of the initial correlation and the range of the interaction in the model. This is discussed in detail in appendix A. In short, one can show for those systems that the exponential bound is negligible for $l_A \gtrsim vt + \xi$ and $l_B \gtrsim vt_0 + \xi$. The derivation assumes that the constants have the same order of magnitude and that the initial support of the observable is small which is a reasonable assumption for a local observable (cf.

appendix A). The total bound can then be optimized, since the minimal value resides on the boundary of the plane of possible values. The minimal bound is then found to be a power-law where the exponent is given by the initial (equilibrium) correlation decay (at $t = t_0 = 0$)

$$|\langle O_A(t) O_B(t_0)\rangle_c| \lesssim \mathscr{C}^{\text{ini}}(L - \nu(t + t_0)) = \frac{\tilde{c}}{\tilde{a}^\beta + (L - \nu(t + t_0))^\beta}. \tag{8}$$

A concrete and simple example is the Luttinger model, which is shown in section 5. In that model the point-like interaction assures that the correlation length $\xi$ is small.

The closed form of the bound allows us to study the build up of correlations in the region away from the light-cone. Inside the light cone, where $L < \nu(t + t_0)$, the correlation function is quasi-unbounded as expected from the Lieb-Robinson bound. Close to the light cone, where $L \sim \nu(t + t_0)$, the light cone is dominated by the reciprocal cut-off value $\tilde{a}^{-\beta}$ with the exponent $\beta$ defined by the equilibrium spatial decay of the correlation function (assuming $\tilde{a} \ll 1$).

It is possible to formally extend the bound in Eq. (7) in a more rigorous mathematical language where one would speak, for example, of a set of vertices that create a lattice and a general metric. In this publication, however, we choose to stick to a simple physical picture. In this picture the locality of the Hamiltonian means that the interaction only couples nearby degrees of freedom which are sitting on a lattice that can be infinite. It is, of course, also possible to apply the results to other spin systems with a more general short-range interaction. As stated above, the requirement is that the considered model fulfills the conditions for the Lieb-Robinson bound as given in Eq. (1).

Analogously to our derivation it is also possible to extend Brayvi *et al.*'s theorem which assumes an exponential decay to non-equal times. This result can be found in section B in the appendix in part B.

The result we derived here can be seen as an extension of different other results presented elsewhere in the literature. We obtain our result using the Lieb-Robinson bound to constrain the support of the time evolution of the operators to a region of a certain width. This width is then fixed to give the tightest bound on the dynamics of the correlation function. The same approach has been used by Brayvi *et al.* in Ref. [36] to constrain the time evolution of the equal-time correlation function for states with exponentially decaying correlations in the initial state. In this particular case the minimization can be achieved analytically obtaining that the dynamics has a light-cone with exponentially decaying leaks outside it. In Sec. B we extend the previous result to non-equal-time correlation, where again it is possible to demonstrate that a sharp light-cone with exponentially decaying leaks is present.

The main point of our analysis is a bound for the dynamics of the correlation function on systems with correlations that are algebraically decaying in the initial state. The equal-time case of this question has already been studied by Kastner in Ref. [38] where the bound is presented in an implicit form. In our work we derive a bound in a different manner and also include non-equal times. Moreover, we provide an explicit bound in the case where the interaction range is smaller than every other length scale. The most important consequence of this analysis is that a light-cone is still present and that the growth of the correlation function near the light-cone is connected to the initial state. In this sense, the initial correlation decay is not completely forgotten over the course of time, but it still influences the dynamics near and outside of the light-cone.

## 3 The Luttinger model

In this section we briefly review the Luttinger model in the bosonization picture. This topic is now standard textbook knowledge and the readers already familiar with that can skip this

section.

An increasing interest in one-dimensional electron models has initially motivated Luttinger to present an interacting fermion model in 1963 [40]. This model resembled the one that Tomonaga had studied 13 years before and which featured the splitting into left- and right-moving excitations for the first time [41]. However, Luttinger assumed a simpler interaction among the fermions and treated the Dirac sea as being infinite [42]. This was corrected two years later by Mattis and Lieb who presented an exact solution for a generalized model by introducing bosonic field operators [43]. They also demonstrated the breakdown of the Fermi liquid theory in one dimension. Further important contributions contained the calculation of one- and two-particle correlation functions [44] and the emergence of a nonuniversal power-law decay. Also, Haldane's studies improved the understanding of this model, most notably that the low-energy behavior implies that the Luttinger model belongs to the newly defined "Tomonaga-Luttinger universality class" [45–47].

In the past decade the interest in the Luttinger model has revived in the context of non-equilibrium dynamics induced by quantum quenches. Cazalilla, for example, has studied the behavior of the model after different quenches [48–50], while Dora and Moessner investigated the out-of-time-ordered density correlators [51].

This work aims at extending the findings of Cazalilla in his extensive study of correlation functions. We focus on investigating the correlation buildup and their behavior in the full space-time plane.

The initial Hamiltonian of interest describes a one-dimensional system with interacting (spinless) electrons and is given in terms of fermion creation and annihilation operators. Since only the low-energy excitations are considered in the following, one assumes that the dispersion relation takes on a linear form. To be able to apply the bosonization machinery, two additional requirements are needed [52]. First, one needs a discrete momentum number $k$ which is achieved by imposing antiperiodic boundary conditions. Second, one extends the dispersion relation to infinity by adding "positron states" which do not have a physical interpretation. In the low-energy regime these states, however, do not contribute. After a complete bosonization procedure the Tomonaga-Luttinger model can be written in terms of bosonic collective excitations which represent a particle-hole pair given by the density operator $\rho_\alpha$. Hereby, $\alpha = L, R$ denotes the left- and right-moving excitations with opposing sign $s_{L/R} = \mp 1$, respectively. Our system of length $L$ is initially prepared as a thermal state with respect to the free theory which only contains a kinetic part

$$H_{\text{kin}} = \int_{-L/2}^{L/2} \frac{\mathrm{d}x}{2\pi} \frac{v_F}{2} : \left[ \rho_L^2(x) + \rho_R^2(x) \right],$$ (9)

with $v_F$ the Fermi velocity. The double dots denote fermionic normal-ordering with respect to a reference state which is chosen to be the Fermi sea [47]. The thermal state is set by a bath at inverse temperature $\beta$ ($\beta = \infty$ describes the ground state). At time $t = 0$ the bath is removed and an interaction quench is performed, i. e. we turn on the interaction parameters $g_2$ and $g_4$ adding the interaction part $H_{\text{int}}$ to the previous Hamiltonian. It is given by

$$H_{\text{int}} = \int_{-L/2}^{L/2} \frac{\mathrm{d}x}{2\pi} : \left[ g_2 \rho_L(x) \rho_R(x) + \frac{g_4}{2} \left( \rho_L^2(x) + \rho_R^2(x) \right) \right].$$ (10)

The interaction is entirely point-like and describes scattering processes within one branch ($g_4$) and the forward scattering between the two branches ($g_2$). Transforming the full Hamiltonian

into momentum space using

$$\rho_\alpha(x) = \frac{N_\alpha}{L} + \mathrm{i}s_\alpha \sqrt{\frac{1}{2\pi L}} \sum_{q>0} e^{-a|q|/2} \sqrt{q}\Big(e^{\mathrm{i}s_\alpha qx} b_{s_\alpha q} - e^{-\mathrm{i}s_\alpha qx} b^\dagger_{s_\alpha q}\Big) \tag{11}$$

yields

$$H = \sum_{q\neq 0} \frac{|q|}{2}\Big[(v_F + g_4)\Big(b^\dagger_q b_q + \frac{1}{2}\Big) + g_2(b^\dagger_q b^\dagger_{-q} + b_q b_{-q})\Big], \tag{12}$$

where several constants which do not affect the dynamics were omitted. To be able to sum over all $q$ we introduced a regularization $\exp(-a|q|/2)$ which is discussed in the next section. One remark to the choice of interactions is necessary: The simplified model with only point-like repulsion bears implications which, however, become irrelevant in the low-energy regime (long-wavelength or $q \sim 0$) [50]. The bosonic operators $b^\dagger_q$ ($b_q$) create (annihilate) a particle-hole excitation with momentum $q$. This Hamiltonian can now be diagonalized with a Bogoliubov rotation given by

$$\begin{pmatrix} b_q \\ b^\dagger_{-q} \end{pmatrix} = \begin{pmatrix} \cosh(\phi) & -\sinh(\phi) \\ -\sinh(\phi) & \cosh(\phi) \end{pmatrix} \begin{pmatrix} B_q \\ B^\dagger_{-q} \end{pmatrix}, \tag{13}$$

with $B_q$ denoting the new bosonic operators in the diagonal basis. The Bogoliubov angle $\phi$ is given by $\tanh(2\phi) = g_2/(v_F + g_4)$. The diagonalized Hamiltonian defines the dispersion relation $\omega(q) = v|q|$ with the renormalized velocity $v = ((v_F + g_4)^2 + g_2^2)^{1/2}$. In the diagonalized form the time-dependence is trivial such that one finds a resulting transformation

$$\begin{pmatrix} b_q(t) \\ b^\dagger_{-q}(t) \end{pmatrix} = \begin{pmatrix} f(q,t) & g^*(q,t) \\ g(q,t) & f^*(q,t) \end{pmatrix} \begin{pmatrix} b_q \\ b^\dagger_{-q} \end{pmatrix}, \tag{14}$$

with $f(q,t) = \cos(v|q|t) - \mathrm{i}\sin(v|q|t)\cosh(2\phi)$ and $g(q,t) = \mathrm{i}\sin(v|q|t)\sinh(2\phi)$.

## 4 Density-density correlation function

In the following we study the density-density correlation function $\langle n(x,t)n(0,t_0)\rangle$ where the expectation value is taken with respect to an initial thermal state (at $T = 0$ the ground state). Separating the density operator in left and right movers $n(x,t) = \rho_L(x,t) + \rho_R(x,t)$, the full density-density operator contains the four contributions

$$\begin{aligned} \langle n(x,t)n(0,t_0)\rangle = &\langle \rho_L(x,t)\rho_L(0,t_0)\rangle + \langle \rho_R(x,t)\rho_R(0,t_0)\rangle \\ &+ \langle \rho_L(x,t)\rho_R(0,t_0)\rangle + \langle \rho_R(x,t)\rho_L(0,t_0)\rangle, \end{aligned} \tag{15}$$

where we omitted terms that oscillate fast with $x$. This corresponds to averaging over $\Delta x \sim (2k_F)^{-1}$ [53]. This averaging procedure erases any non-universal power-law decay that is non-Fermi liquid like. Similar quantities like Eq. (15) have already been studied in the literature, however the full calculation at non-equal time has not been done to give a closed-form expression [49–51]. In Ref. [50] the authors discussed the sum which appears in the calculation in the infinite-time limit without performing it.

The calculation of the full density-density correlation function requires plugging Eq. (11) and (14) in (15) which results in four terms. In order to be able to sum over $q$, a regularization $\exp(-|q|a/2)$ defined by the "effective bandwidth" $a^{-1}$ is introduced [46]. One usually considers the case where $a \to 0^+$, however a small but finite value is closer to the real world, since it constrains possible scattering events. It will become clear in the following that this basically only affects the region around the light cone, where a finite $a$ limits the peak height. Away from it one can safely let $a \to 0^+$.

## 4.1 Zero temperature

One finds at $T = 0$ for a finite system size $L$ that

$$
\langle n(x,t)n(0,t_0)\rangle_{T=0} = \frac{1}{4\pi^2}\Big[ \big(\gamma\sqrt{1+\gamma^2} - \gamma^2 - 1\big)\big(U(x+v(t-t_0),a) + U(x-v(t-t_0),a)\big)
$$
$$
- \big(\gamma\sqrt{1+\gamma^2} - \gamma^2\big)\big(U(x+v(t+t_0),a) + U(x-v(t+t_0),a)\big)
$$
$$
- \mathrm{i}\big(\sqrt{1+\gamma^2} - \gamma\big)\big(V(x+v(t-t_0),a) - V(x-v(t-t_0),a)\big)\Big],
$$

(16)

with

$$
U(\tau,a) = \frac{2\pi^2}{L^2}\frac{1 - \cosh\left(\frac{2\pi}{L}a\right)\cos\left(\frac{2\pi}{L}\tau\right)}{\left(\cosh\left(\frac{2\pi}{L}a\right) - \cos\left(\frac{2\pi}{L}\tau\right)\right)^2}
$$

(17)

$$
V(\tau,a) = \frac{2\pi^2}{L^2}\frac{\sinh\left(\frac{2\pi}{L}a\right)\sin\left(\frac{2\pi}{L}\tau\right)}{\left(\cosh\left(\frac{2\pi}{L}a\right) - \cos\left(\frac{2\pi}{L}\tau\right)\right)^2}.
$$

(18)

$\gamma = g_2/v$ denotes the quench parameter which causes the time-dependence (since the $g_4$ term commutes with the non-interacting Hamiltonian). We will now sometimes use the notation $\tau$ for any combination of $x \pm v(t \pm t_0)$.

To discuss the functions $U(\tau,a)$ and $V(\tau,a)$ we naively let $a \to 0^+$. This limiting produce gives the known result where $U(\tau) = d(\tau|L)^{-2}$, where $d(\tau|L) = L\sin(\pi x/L)/\pi$ is the conformal distance which becomes $d(\tau) = \tau$ in the thermodynamic limit [49]. The imaginary part of Eq. (16) converges to a sum of derivatives of $\delta$-distributions on the left- and right-moving light cone (where $\tau = 0$), because $V(\tau) = -\pi\partial_\tau\delta(\tau)$ and consequently only contributes there.

However, considering $a$ to be very small, but non-zero, changes the behavior close to the light cone. In this case one needs to approximate $U(\tau,a)$ and $V(\tau,a)$ for small $\tau$ and $a$ and finds $U(\tau \approx 0) \approx (\tau^2 - a^2)/(\tau^2 + a^2)^2$ with $U(\tau = 0, a) = -1/a^2$ on the light cone and $V(\tau \approx 0) \approx 2a\tau/(\tau^2 + a^2)^2$ with $S(\tau = 0, a) = 0$. The arising divergences create two light cones in both directions (we will consider $x > 0$ from now on): an outer light cone at $x - v(t+t_0) = 0$ and an inner light cone at $x - v(t-t_0) = 0$. The first thing to note is that the behavior close to a light cone is independent of the system size (it is identical to the result in the thermodynamic limit). While $U(\tau,a)$ is symmetrically algebraically peaked with a maximum value $-1/a^2$, the width of the $V(\tau,a)$-double peak depends strongly on $a$. From its functional form one can see that $V(\tau,a)$ is pointsymmetric with respect to $\tau = 0$: Considering the peak around a right-moving light cone ($\tau = x - v(t+t_0) = 0$) the time-like region is defined by $x < v(t+t_0)$ or $\tau < 0$, while the space-like region is given by $\tau > 0$. Thus, it has a negative peak inside the light cone, while the outside peak is positive. For a left-moving light cone it is $\tau > 0$ ($\tau < 0$) for the time-like (space-like) region, but the negative sign in front of the corresponding function $V(x + v(t+t_0))$ assures the same behavior holds for both light cones (see also Ref. [51]).

Finally, one can recover the result by Cazalilla [49], if one considers only the right-moving density-density correlation at equal times ($t = t_0$). This requires neglecting the terms proportional to $\gamma\sqrt{1+\gamma^2}$, since these terms derive from $L-R$ and $R-L$ contributions. Moreover, one needs an additional factor of $1/2$, since both the $L-L$ and $R-R$ are identical in this case.

The function that describes the *initial* correlations can be found by setting $t = t_0 = 0$ in Eq. (16). These correlations are those that were present *before* the time-evolution. They are

given by

$$\langle n(x,0)n(0,0)\rangle^{\text{ini}}_{T=0} = -\frac{1}{2\pi^2}\frac{x^2-a^2}{(x^2+a^2)^2}. \tag{19}$$

For $x \gg a$ it is approximately $\langle n(x,0)n(0,0)\rangle^{\text{ini}}_{T=0} \approx -1/(2\pi^2 x^2)$. Generally, we call the correlations that are *not* $\gamma$-dependent *pre-quench* correlations (also indicated by "pq"). For equal-time correlators ($t = t_0$) these contributions are identical to the initial correlations, however, if $t \neq t_0$ the different measurement times change the correlation function. In the space-like region (outside of the light cone) where $x \gg v(t+t_0) > v(t-t_0) \gg a$, however, the difference is negligible, because

$$\langle n(x,t)n(0,t_0)\rangle^{\text{pq}}_{T=0} = -\frac{1}{2\pi^2 x^2}\left(1 + 3\left(\frac{v(t-t_0)}{x}\right)^2\right) \approx -\frac{1}{2\pi^2 x^2}. \tag{20}$$

The third part of Eq. (16) are the *quench-induced* contributions (indicated by "qi") which depend on $\gamma$. They are thoroughly discussed in section 6.

## 4.2 Non-zero temperature

In the following we extend the previous results to non-zero temperature where the initial state is a thermal state at inverse temperature $\beta$. At $T = 0$ ($\beta = \infty$) the contractions $\langle b_q b_q^\dagger\rangle = 1 + \langle b_q^\dagger b_q\rangle = 1$, because the Bose-Einstein distribution $\langle b_q^\dagger b_q\rangle = 1/(\exp(\beta v_F |q|)-1)$ vanishes. At $T > 0$ additional terms add to the zero temperature result. The full result for a finite system of size $L$ can be found in the appendix in section D. We now focus on the result in the thermodynamic limit, where one can execute the infinite sum and gets:

$$
\begin{aligned}
\langle n(x,t)n(0,t_0)\rangle_{\text{therm}} = {}& \langle n(x,t)n(0,t_0)\rangle_{T=0} - \frac{\gamma\sqrt{1+\gamma^2}-\gamma^2-1}{(2\pi\beta v_F)^2} \\
&\times \Bigg[ \text{DiG}_1\left(1 + \frac{a}{\beta v_F} - \frac{\mathrm{i}}{\beta v_F}(x + v(t-t_0))\right) + \text{DiG}_1\left(1 + \frac{a}{\beta v_F} + \frac{\mathrm{i}}{\beta v_F}(x + v(t-t_0))\right) \\
&+ \text{DiG}_1\left(1 + \frac{a}{\beta v_F} - \frac{\mathrm{i}}{\beta v_F}(x - v(t-t_0))\right) + \text{DiG}_1\left(1 + \frac{a}{\beta v_F} + \frac{\mathrm{i}}{\beta v_F}(x - v(t-t_0))\right) \Bigg] \\
&+ \frac{\gamma\sqrt{1+\gamma^2}-\gamma^2}{(2\pi\beta v_F)^2} \\
&\times \Bigg[ \text{DiG}_1\left(1 + \frac{a}{\beta v_F} - \frac{\mathrm{i}}{\beta v_F}(x + v(t+t_0))\right) + \text{DiG}_1\left(1 + \frac{a}{\beta v_F} + \frac{\mathrm{i}}{\beta v_F}(x + v(t+t_0))\right) \\
&+ \text{DiG}_1\left(1 + \frac{a}{\beta v_F} - \frac{\mathrm{i}}{\beta v_F}(x - v(t+t_0))\right) + \text{DiG}_1\left(1 + \frac{a}{\beta v_F} + \frac{\mathrm{i}}{\beta v_F}(x - v(t+t_0))\right) \Bigg],
\end{aligned}
\tag{21}
$$

with the derivative of the DiGamma-function $\text{DiG}_1(z) = \frac{\mathrm{d}}{\mathrm{d}z}\ln\Gamma(z)$. This general result is new and has not been calculated before. The behavior close to the light cone is dominated by the sharp peaks of the $T = 0$ case, since the DiGamma-functions only give a constant contribution there ($2\,\text{DiG}_1(1) = \pi^2/3$).

In the space-like region from the light cone the limit $a \to 0^+$ can be taken and the additional terms can be summarized to $\text{DiG}_1(1 - \mathrm{i}\tau/(\beta v_F)) + \text{DiG}_1(1 + \mathrm{i}\tau/(\beta v_F)) = (\beta v_F)^2/\tau^2 - \pi^2\sinh^{-2}(\pi\tau/(\beta v_F))$. The algebraic decay terms now exactly cancel the $T = 0$ terms such that it finally becomes

$$\langle n(x,t)n(0,t_0)\rangle_{\text{therm}} =$$
$$\frac{1}{4\pi^2\xi^2}(\gamma\sqrt{1+\gamma^2}-\gamma^2-1)\left[\sinh^{-2}\left(\frac{x+v(t-t_0)}{\xi}\right)+\sinh^{-2}\left(\frac{x-v(t-t_0)}{\xi}\right)\right]$$
$$-\frac{1}{4\pi^2\xi^2}(\gamma\sqrt{1+\gamma^2}-\gamma^2)\left[\sinh^{-2}\left(\frac{x+v(t+t_0)}{\xi}\right)+\sinh^{-2}\left(\frac{x-v(t+t_0)}{\xi}\right)\right]$$
$$-\frac{i}{4\pi}(\sqrt{1+\gamma^2}-\gamma)\left[\partial_{(x-v(t-t_0))}\delta(x-v(t-t_0))-\partial_{(x+v(t-t_0))}\delta(x+v(t-t_0))\right],$$
$$\tag{22}$$

where the temperature induced correlation length $\xi = \beta v_F/\pi$ has been introduced. Another form of writing the expression above is taking the $T = 0$ result and substituting the conformal distance $d(\tau)$ with the newly defined (thermal) distance $d(\tau,\beta) := \xi\sinh(\tau/\xi)$ as hinted at in Ref. [49]. This is, of course, only true in the thermodynamic limit or if the correlation length scale is way shorter than the system size ($\xi = \beta v_F/\pi \ll L$).

The thermal correlation length $\xi$ now separates two different regimes: if $x \pm v(t \pm t_0) \ll \xi$ we observe again the algebraic decay regime as in the $T = 0$ case which is now followed by an exponentially decaying regime for $x \pm v(t \pm t_0) \gg \xi$. Both regimes will be discussed in the form of the anticommutators in section 6.

## 5 Explicit bound for the Luttinger model

In this section we explicitly verify the bound presented in section 2 for the two observables $O_A = n(x)$ and $O_B = n(0)$ at non-equal times $t$ and $t_0$. The compact support contains only the spatial point $B = 0$ and $A = x$, respectively. To do this we define a function $\mathscr{C}^{\text{ini}}(x)$ which describes the absolute value of the initial correlations

$$\mathscr{C}^{\text{ini}}(x) := |\langle n(x,0)n(0,0)\rangle_c|. \tag{23}$$

At $t = t_0 = 0$ and zero temperature we know the exact expression for the Luttinger model from Eq. (16):

$$\mathscr{C}^{\text{ini}}(x) = \frac{1}{2\pi^2}\frac{|x^2-a^2|}{(x^2+a^2)^2} \le \frac{1}{2\pi^2}\frac{1}{x^2+a^2}. \tag{24}$$

Here, we used that $\langle n(x)\rangle = 0$. As discussed earlier $\mathscr{C}^{\text{ini}}(x)$ describes a power-law decay with an exponent $\beta = 2$, while $\tilde{c} = 1/2\pi^2$. From the approximated bound in Eq. 8 we thus expect a bound which has the same exponent.

We prove that this is indeed fulfilled by bounding the full density-density correlation function for $t > t_0 > 0$. Although we have calculated the exact correlation function also for the inside of the two light cones in the previous sections, Eq. (8) states that it is quasi-unbounded inside of the outer light cone. We therefore only focus on the decay outside of the outer light cone plus a small offset, i. e. where $x \ge v(t+t_0)+\sqrt{3}a$. The small shift by $\sqrt{3}a$ is necessary to perform the following estimation (for a more detailed explanation see section C in the appendix):

$$\frac{(x+v(t+t_0))^2-a^2}{((x+v(t+t_0))^2+a^2)^2} \le \frac{(x+v(t-t_0))^2-a^2}{((x+v(t-t_0))^2+a^2)^2}$$
$$\le \frac{(x-v(t-t_0)))^2-a^2}{((x-v(t-t_0))^2+a^2)^2} \le \frac{(x-v(t+t_0)))^2-a^2}{((x-v(t+t_0))^2+a^2)^2}. \tag{25}$$

The bounds are tight if $t = t_0 = 0$. Then, noticing that all terms have the same sign, one bounds each term via Eq. (25) and finds that

$$
\begin{aligned}
|\langle n(x,t)n(0,t_0)\rangle_c| &= \frac{1}{4\pi^2}\left|\left(\gamma\sqrt{1+\gamma^2}-\gamma^2-1\right)\right.\\
&\quad \times\left(\frac{(x-v(t-t_0))^2-a^2}{((x-v(t-t_0))^2+a^2)^2}+\frac{(x+v(t-t_0))^2-a^2}{((x+v(t-t_0))^2+a^2)^2}\right)\\
&\quad \left.-\left(\gamma\sqrt{1+\gamma^2}-\gamma^2\right)\left(\frac{(x+v(t+t_0))^2-a^2}{((x+v(t+t_0))^2+a^2)^2}+\frac{(x-v(t+t_0))^2-a^2}{((x-v(t+t_0))^2+a^2)^2}\right)\right|\\
&\leq \frac{1}{4\pi^2}\left|\left(\gamma\sqrt{1+\gamma^2}-\gamma^2-1\right)\left(\frac{(x-v(t+t_0))^2-a^2}{((x-v(t+t_0))^2+a^2)^2}+\frac{(x-v(t+t_0))^2-a^2}{((x-v(t+t_0))^2+a^2)^2}\right)\right.\\
&\quad \left.-\left(\gamma\sqrt{1+\gamma^2}-\gamma^2\right)\left(\frac{(x-v(t+t_0))^2-a^2}{((x-v(t+t_0))^2+a^2)^2}+\frac{(x-v(t+t_0))^2-a^2}{((x-v(t+t_0))^2+a^2)^2}\right)\right|\\
&= \frac{1}{4\pi^2}\left|(-2)\frac{(x-v(t+t_0))^2-a^2}{((x-v(t+t_0))^2+a^2)^2}\right|\\
&= \frac{1}{2\pi^2}\frac{|(x-v(t+t_0))^2-a^2|}{((x-v(t+t_0))^2+a^2)^2} = \mathscr{C}^{\text{ini}}(x-v(t+t_0)).
\end{aligned}
\tag{26}
$$

Analogously to the initial case, there is again the upper limit of Eq. (26) with $\mathscr{C}^{\text{ini}}(x - v(t + t_0)) \leq 1/(2\pi^2)/((x - v(t + t_0))^2 + a^2)$ that takes on the form as given in Eq. (8).

Thus, we have explicitly demonstrated that in the Luttinger model the correlations outside of the outer light cone can be bounded by the initial correlations shifted by the current position of the light cone. In a physical picture one can think of the correlation as being "dragged" with the quasiparticles which move with the light cone speed. As soon as any quasiparticle has passed a region the correlation function is basically unbounded. In a real system the bound does not need to be tight, but the real expression can fall off even quicker (see discussion of the exact decay in section 6).

# 6 The spatiotemporal correlation build-up

In this section we want to study the dynamics of the anti-commutator and commutator of the density-density correlation function that have been calculated in part 4 at zero and non-zero temperatures. Hence, we define the functions

$$
C_\pm(x,t,t_0) := \langle[n(x,t),n(0,t_0)]_\pm\rangle_{\text{therm,}}
\tag{27}
$$

where $[\cdot,\cdot]_\pm$ denotes the anticommutator or commutator. Since the anticommutator is just twice the real part of the density-density correlation and the commutator the corresponding doubled imaginary part of Eqs. (16) and (21), one immediately finds the resulting expressions. The expectation value is taken with respect to a thermal state defined by the pre-quench Hamiltonian at inverse temperature $\beta$ via the distribution $\exp(-\beta H_{\text{kin}})/Z$ ($\beta = \infty$ corresponds to the ground state). Hereby, we consider the thermodynamic limit, since we focus on the decay regimes and the finite size effects have been discussed elsewhere [49]. In the following, when we study the anticommutator we often subtract the initial contributions $C_+^{\text{ini}}(x) := C_+(x,0,0)$. This is done, because we want to study how the correlations buildup without having the background of the equilibrium (if we do include it, it is mentioned).

## 6.1 Analysis of the commutator $C_-(x, t, t_0)$

As has been discussed before [50, 51], the commutator is just a complex number (independent of the inverse temperature $\beta$) and as such it is independent of the initial state (also compare Eq. (16) and (22)). Due to the very sharp $\delta$-distribution form of the commutator, it becomes clear that the only contribution results from the left- and right-moving light cones where $x \pm v(t - t_0) = 0$. Physically, this can be understood by regarding the Luttinger model as being effectively a relativistic theory with a well-defined speed of light $v$. The linear dispersion creates an identical ballistic propagation for all quasiparticles (particle-hole excitations) with exactly this speed (no curvature in the dispersion). Therefore, there is also no signal at equal times when $t = t_0$. Generally, it follows that we do not observe any "leak" outside the light cone and one sees that the LR-bounds are trivially fulfilled. Not taking the limit $a \to 0^+$ broadens the peak to a Lorentzian form with FWHM$= 2a$. However, the arising contributions outside the light cone do not violate the LR-bounds.

## 6.2 Analysis of the anticommutator $C_+(x, t, t_0)$

This simple behavior can, of course, not be seen when studying the anticommutator which is not bounded by the Lieb-Robinson bounds without further restrictions (see derivation of a bound in part 2). While the commutator is independent of the initial state (it is an operator norm), the anticommutator strongly depends on it. In this sense, the anticommutator has to be understood as a statistical property, since one needs to perform many weak measurements to determine its exact shape.

The equal-time ($t = t_0$) anticommutator has already been calculated in Ref. [49], but with a focus on the long-time convergence to a generalized Gibbs ensemble and not on the spatiotemporal correlation buildup.

The analysis starts with the $T = 0$ case, where the anticommutator is defined by twice the real part of Eq. (16). Moreover, we simplify by considering $a \to 0^+$ and taking the thermodynamic limit $L \to \infty$. This can be done, because a finite $a$ is only needed in the vicinity of the light cone where $\tau \approx 0$. If a finite $a$ is required, we will from now on explicitly discuss the change. In order to study the correlation buildup induced by the interaction quench we subtracted the initial correlations (if $t = t_0$, it is initial = pre-quench correlations) which can be found by setting $t = 0$ in Eq. (16):

$$C_{+,\text{alg}}^{\text{ini}}(x) = -\frac{1}{\pi^2 x^2}. \tag{28}$$

They show the usual Fermi gas or free fermion behavior, since initially the model is non-interacting. Having subtracted this contribution the quench-induced part of the anticommutator reads

$$C_+(x, t) = -\frac{1}{2\pi^2}(\gamma \sqrt{1 + \gamma^2} - \gamma^2) \times \left( \frac{1}{(x - 2vt)^2} + \frac{1}{(x + 2vt)^2} - \frac{2}{x^2} \right) =$$

$$= (\gamma \sqrt{1 + \gamma^2} - \gamma^2) \times \left( \frac{1}{2(1 - 2vt/x)^2} + \frac{1}{2(1 + 2vt/x)^2} - 1 \right) C_{+,\text{alg}}^{\text{ini}}(x). \tag{29}$$

In the last expression we have expressed the anticommutator in terms of the initial correlations. From now on, we will always use this description to be able to relate the quench-induced correlations to the initial ones. The absolute value of this function $|C_+(x, t)|$ is depicted in Fig. 2a for the right-moving light cone where the initial correlations have been subtracted. In a physical picture the figure shows the correlations which are induced by the movement of the quasiparticles after the quench [18]. In the figure one sees that for a fixed $x$ the anticommutator is zero initially, but then growing to the sharply peaked light cone on the diagonal with

time. This corresponds to a vertical cut through the plot. This behavior can be understood as follows: At $t = 0$ particle-hole excitations get excited by the global quench which then spread in both directions with velocity $2v$ and possibly correlate all regions that they have passed. This is called the correlation buildup. Since all quasiparticles move at the same speed, they will continue their coherent movement forever. Thus, there is a huge correlation peak (named light cone) that indicates that all excitations come from the same point $x = 0$ (at $t = 0$) to which they are therefore highly correlated. In order to discuss the growth rate and the decay regime outside of the light cone one needs to approximate Eq. (29) in the space-like region, i.e. where $x \gg 2vt$. This results in

$$C_+^{\text{sl}}(x, t) = 3\left(\frac{2vt}{x}\right)^2 \left(\gamma\sqrt{1+\gamma^2} - \gamma^2\right) C_{+,\text{alg}}^{\text{ini}}(x). \tag{30}$$

In this region, one finds that the growth of correlations is proportional to $t^2/x^4$ resulting in algebraically increasing "leaks" outside the light cone if one considers a fixed $x$ (vertical cut of Fig. 2a). Closer to the light cone, $x \sim 2vt$, the full algebraic growth $\propto 1/(x-2vt)^2$ takes over before the light cone peaks to $-1/a^2$ (or infinitely high if there is an infinite number of excitations, i.e. $a \to 0^+$). While the correlation function grows with time for a fixed $x$, the behavior is different outside the light cone for a fixed time $t$: As has been shown in section 2 the decay outside the light cone can be effectively bounded by the initial (equilibrium) decay that moves with the speed of the quasiparticles. Only the amplitudes might change. In the Luttinger liquid we notice in section 5 that the decay is only tightly bounded in the close vicinity of the light cone. In this sense, the bound we derived in section 2 can be interpreted as the initial correlations being "dragged" with the quasiparticles during the time evolution. Mathematically, this is given by the simple substitution $x \to x - v(t + t_0)$ in the correlation function. In the very deep space-like region where $x \gg 2vt$, however, the correlation function decays faster with $1/x^4$. This is depicted in Fig. 2b which cuts Fig. 2a horizontally for a fixed time.

Finally, we discuss the structure that is visible inside the light cone. There we can identify a second signal where correlations vanish (see Fig. 2b). There, the time-independent stationary, or time-like, ($x \ll 2vt$) correlations $C_{+,\text{alg}}^{\text{tl}}(x) = -(\gamma\sqrt{1+\gamma^2} - \gamma^2)C_{+,\text{alg}}^{\text{ini}}(x)$ cancel the term coming from the light cone. In other words, it is exactly at $x = (2vt)/\sqrt{3}$ where $C_+(x, t) = 0$. Since the time-like decay is stationary, it is also the remaining contribution in the infinite-time limit. Adding the initial correlation again one notices that the initial decay exponent of the non-interacting Fermi gas prevails, but with a reduced strength $-(\gamma\sqrt{1+\gamma^2} - \gamma^2) + 1$ which is bounded by $1/2$ from below. In this sense, we have an interacting Fermi liquid. This also confirms our expectation to not see any non-universal power-law, since we averaged over $\Delta x \sim (2k_F)^{-1}$ [53].

There is an interesting difference to results in the literature where only the $R - R$ density-density correlation has been studied [49]. There, the infinite-time limit is always larger than the initial correlations (for a repulsive interaction). However, taking the full physical fermion density into account shows the physically expected behavior: the interaction limits the movement of the quasiparticles and thus reduces the initial correlations. The exponent of the decay, however, remains the same.

The previous results drastically change when one considers a thermal initial state. In that case, one needs to analyze the real-part of Eq. (22) (with $a \to 0^+$). The initial correlation decay is found to be $C_+^{\text{ini}}(x) = -1/(\pi\xi\sinh(x/\xi))^2$. We depict the full spatiotemporal behavior without the initial contributions in Fig. 3a. Analyzing the defining equation one now observes two different regimes which are governed by the temperature via the new correlation length $\xi$. If $x \ll \xi$ one finds the algebraic regime as in Eq. (28). Physically, in this regime we see

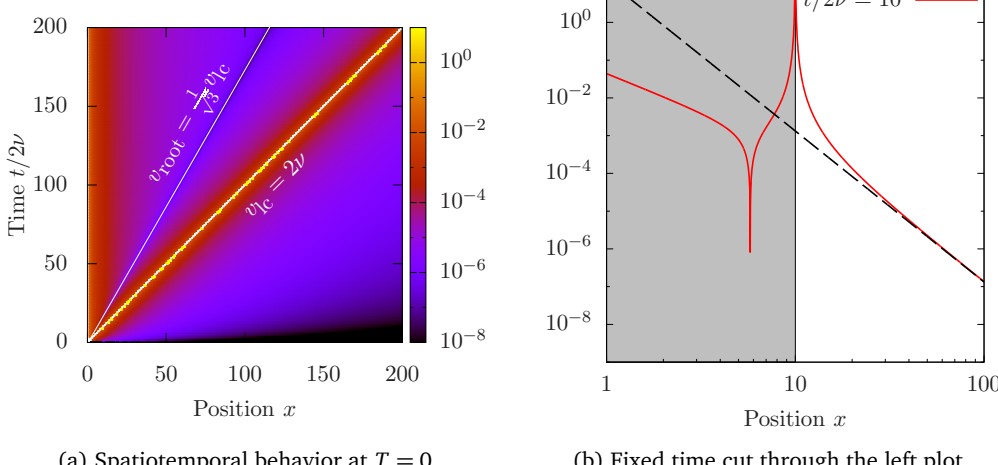

(a) Spatiotemporal behavior at $T = 0$  (b) Fixed time cut through the left plot

Figure 2: Spatiotemporal behavior of the equal-time anticommutator $|C_+(x,t)|$ at $T = 0$ ($\beta = \infty$) without the initial correlations (as given in Eq. (29)). The signal on the diagonal is the light cone with an algebraic (power-law-like) decay. A horizontal cut through the plot for time $t/2\nu = 10$ is shown on the right (Fig. 2b) with the algebraic decay of Eq. (30) indicated by the black dashed line. The grey shaded area marks the inside of the light cone with a stationary algebraic correlation decay starting at $x = 0$. Due to different signs this contribution cancels the inner decay coming from the light cone leading to an inner structure with vanishing correlation at $x_{\text{root}} = t/(2\nu\sqrt{3})$.

length scales much shorter than the one induced by the temperature. However, for $x \gg \xi$ the decay becomes exponential with

$$C_{+,\exp}^{\text{ini}}(x) = -\frac{4}{\pi^2 \xi^2} e^{-2x/\xi}. \tag{31}$$

In summary, the temperature introduces a new correlation length which separates two different decay regimes. This behavior of the time-independent initial correlation has already been described in Ref. [53]. We now show that it extends to the quench-induced correlations by studying the full dynamics. We start with the correlation buildup far away from any causal interference, i.e. the space-like region ($x \gg 2\nu t$) where we see that the two initial decay regimes persist in the full spatiotemporal plane. Closer to the light cone, where $|x \pm 2\nu t| < \xi$, we observe an algebraic decay region with similar expressions for the space-like and time-like case as the ones found for $T = 0$. Outside of this region, however, we find an exponential decay of correlations which has been observed in many other systems. The space-like expression for this region is

$$C_{+,\exp}^{\text{sl}}(x,t) = 2\left(\frac{2\nu t}{\xi}\right)^2 \left(\gamma\sqrt{1+\gamma^2} - \gamma^2\right) C_{+,\exp}^{\text{ini}}(x), \tag{32}$$

where the initial correlation decay is now given by Eq. (31). It looks similar to the algebraic version (Eq. (28)), but with a different prefactor, a different decay of the initial correlations (exponential instead of algebraic) and $\xi$ takes over the role of $x$ in the buildup process which is again proportional to $t^2$.

In the time-like region ($x \ll 2\nu t$) one finds the same algebraic decay close to $x = 0$ as for

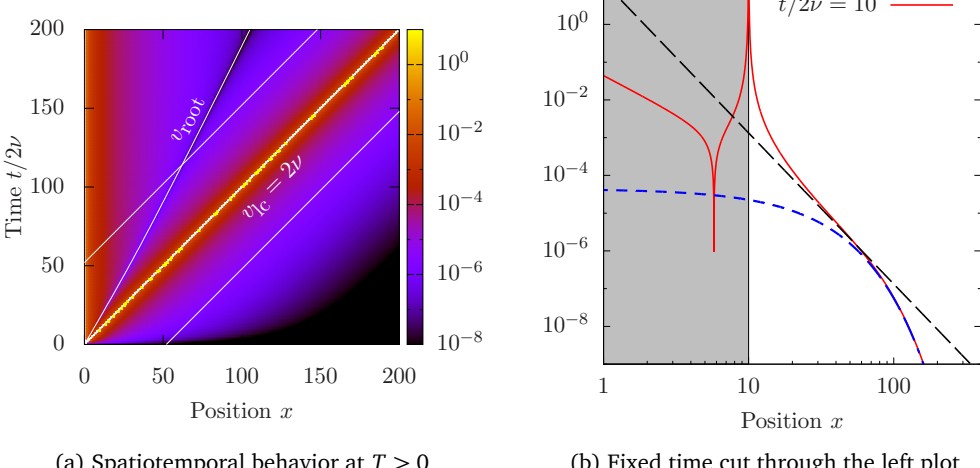

(a) Spatiotemporal behavior at $T > 0$      (b) Fixed time cut through the left plot

Figure 3: Spatiotemporal behavior of the equal-time anticommutator $|C_+(x,t)|$ for $\beta = 30$ as given by Eq. (22) without the intial contribution. The Fermi velocity is chosen such that $\xi = \beta = 30$. The signal on the diagonal is the light cone with an algebraic (power-law-like) decay for $x < \sqrt{3}\xi$ followed by an exponential decay. The algebraic regime is found between the white lines. A horizontal cut through the plot for time $t/2\nu = 10$ is shown on the right (Fig. 3b) with both decay regimes. The black dashed line shows the algebraic regime as in the $T = 0$ case, whereas the blue dashed line represents the exponential decay given by Eq. (32). The size of the algebraic decay regimes shrinks with increasing temperature. The grey shaded area marks the inside of the light cone with a stationary algebraic correlation decay starting at $x = 0$. Due to different signs this contribution cancels the inner decay coming from the light cone leading to an inner structure with vanishing correlation at $x_{\text{root}}$ (see text for formula).

$T = 0$, followed by an exponential decay rate $C_{+,\exp}^{\text{tl}}(x,t) = -(\gamma\sqrt{1+\gamma^2} - \gamma^2)C_{+,\exp}^{\text{ini}}(x)$. The point where the anticommutator vanishes is now given by

$$x_{\text{root}} = \xi \cosh^{-1}\left(\frac{1}{2}\sqrt{1 + \sqrt{5 + 4\cosh(4\nu t/\xi)}}\right). \tag{33}$$

In conclusion, one finds that the algebraic regime present around the peaks on the light cones at $T = 0$ shrinks with increasing temperature. It is followed by a new, temperature induced exponential decay. Both regimes reflect the initial correlations and can be seen in the $\log-\log$-plot in Fig. 3a and 3b.

We now switch to the non-equal-time anticommutator where $t \neq t_0$. We can write $C_+(x,t,t_0) = C(x, t-t_0, t+t_0)$, i.e. it only depends on the difference and sum of the two times. We can therefore write the expressions in terms of time differences $t - t_0$ and find that $t + t_0 = t - t_0 + 2t_0$ where $t_0$ is fixed. Fig. 4a shows the entire spatiotemporal plane of the anticommutator $C_+(x, t-t_0)$. We show the positive time axis which corresponds to values $t > t_0$. Physically, this means that we quench the system at time $t = 0$, measure the density of excitations after time $t_0$ at $x = 0$ and again after time $t > t_0$ at position $x$. The mentioned figure shows the expected behavior: the quench creates excitations that all move with the renormalized Fermi velocity $2\nu$ in both directions. At the first measurement at time

$t_0$ they have propagated by $2vt_0$. Now we measure again at a different position $x$ after an additional time $t - t_0$. We find two distinct signals: One describes the outer light cone at $x = v(t + t_0) = v(t - t_0 + 2t_0)$ which represents the spread of quasiparticles due to the quench until the first measurement after $t_0$. The other one is the light cone on the diagonal, i.e. at $x = v(t - t_0)$ and shows the correlation of the first measurement to the second measurement. Since both light cones have different signs, we again see a light cone of vanishing correlations in between the peaks. This double light cone structure has also been observed in other models and appears, because the Luttinger model with its well-defined propagation speed is effectively a relativistic theory [17]. In the following we want to express the corresponding decay regimes in terms of the pre-quench correlation decay. The pre-quench correlations are those that we would see without quenching ($\gamma = 0$) for $x \gg v(t - t_0)$. They read

$$C_+^{\mathrm{pq}}(x, t, t_0) = -\frac{1}{\pi^2 x^2}\left(1 + 3\left(\frac{v(t - t_0)}{x}\right)^2\right) = C_{+,\mathrm{alg}}^{\mathrm{ini}}(x)\left(1 + 3\left(\frac{v(t - t_0)}{x}\right)^2\right). \quad (34)$$

The leading (slowest) decay term is identical to the initial correlation decay at equal times in Eq. (28). The next order expansion term reflects the change of the correlation due to different times and is negligible if one considers a point far away from the light cone. Thus, we will approximate $C_+^{\mathrm{pq}}(x, t, t_0) \approx C_+^{\mathrm{ini}}(x)$. The decay outside of the outer light cone (space-like region) shows a behavior closely related to the equal-time correlation function. At $T > 0$ the algebraic decay

$$C_{+,\mathrm{alg}}^{\mathrm{sl}}(x, t, t_0) = \frac{12 v^2 t t_0}{x^2}\left(\gamma\sqrt{1 + \gamma^2} - \gamma^2\right) C_{+,\mathrm{alg}}^{\mathrm{ini}}(x), \quad (35)$$

which is the only decay at $T = 0$ is enclosed by an exponential decay

$$C_{+,\mathrm{exp}}^{\mathrm{sl}}(x, t, t_0) = \frac{8 v^2 t t_0}{\xi^2}\left(\gamma\sqrt{1 + \gamma^2} - \gamma^2\right) C_{+,\mathrm{exp}}^{\mathrm{ini}}(x), \quad (36)$$

similar to the equal-time case. In the time-like region one finds that the quench-induced correlations decay proportional to $1/t^2$, since

$$C_{+,\mathrm{alg}}^{\mathrm{tl}}(x, t, t_0) = \frac{\gamma\sqrt{1 + \gamma^2} - \gamma^2}{\pi^2 v^2}\left[\frac{1}{(t - t_0)^2} - \frac{1}{(t + t_0)^2} + 3\frac{x^2}{v^2}\left(\frac{1}{(t - t_0)^4} - \frac{1}{(t + t_0)^4}\right)\right]. \quad (37)$$

Hence, $C_{+,\mathrm{alg}}^{\mathrm{tl}}(x, t, t_0)$ vanishes in the infinite-time limit and only the initial correlations remain. However, for a finite time difference $t - t_0$ Eq. (37) demonstrates that the spatial decay inside of the light cone is independent of $x$ in the first order approximation (see Fig. 4b).

# 7 Conclusions

In this paper we discussed the spreading of local observables following a quantum quench in many-body quantum systems. In particular, we elucidated the difference between the commutator and anticommutator and the influence of the initial state on the dynamics.
First, we derived a bound for the expectation value of local observables at non-equal times for systems with algebraically decaying spatial correlations in the initial state. This result is obtained using the Lieb-Robinson bound and generalizes the findings by Brayvi et al. [36] and Kastner [38] to non-equal time correlations. By optimizing the bound we demonstrated that it effectively describes a light cone which velocity is the Lieb-Robinson velocity. Outside the light cone, correlations are non-zero but algebraically suppressed with a power law determined

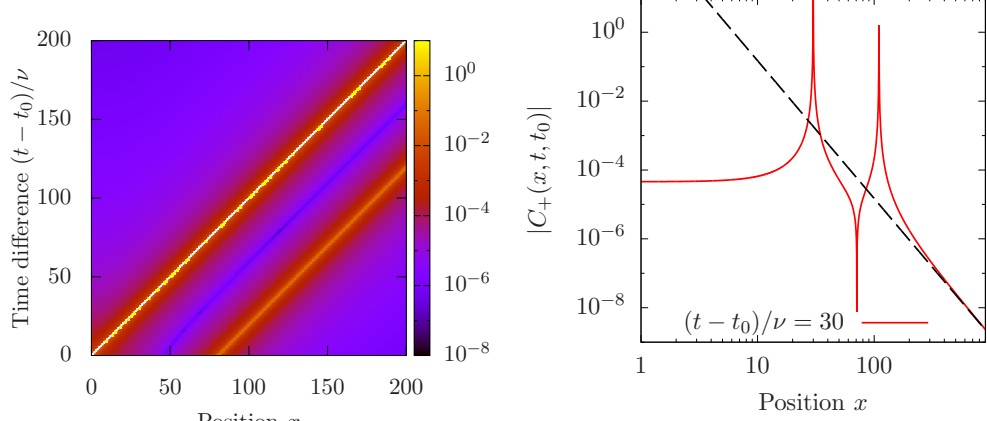

(a) Spatiotemporal behavior at $T = 0$ and $t \neq t_0 > 0$      (b) Fixed time cut through the left plot

Figure 4: Spatiotemporal behavior of the non-equal-time anticommutator $|C_+(x, t, t_0)|$ for $T = 0$ and $t_0/\nu = 40$ without the initial contributions. The anticommutator takes on the form of a double light cone: the signal on the diagonal is the stationary decay which propagates further if $t > t_0$. A horizontal cut through the plot for time $(t - t_0)/\nu = 30$ is shown on the right (Fig. 4b). Due to different signs the two light cone contributions at $(t - t_0)\nu = 30$ and $(t + t_0)/\nu = 110$ cancel exactly in the middle of the peaks. We again see an algebraic decay regime outside of the outer light cone. This is directly related to the initial correlations (see text).

by the equilibrium (initial) correlations. Moreover, the same power law determines the fast buildup of correlations close to the light cone, where the maximum occurs. The dependence on the initial state reflects the difference of the bounds: whereas the LR-bound is an upper bound on the operator norm and thus completely independent of the initial state, the bound for correlation functions depends non-trivially on the initial state.

In a next step, we verified the bound by studying the time evolution of an exactly solvable model: the Luttinger liquid. We investigated the dynamics induced by a sudden switch on of the interactions starting from different initial states: the ground state, which has algebraically decaying correlations, and a thermal mixture, where correlations decay exponentially at long distances.

Using bosonization techniques, we derived exact expressions for the density-density correlation at different positions and times. We discussed both the commutator and the anticommutator of the mentioned observables in the full spatiotemporal plane. While the commutator gives a simple delta function on the light cone and as such trivially fulfills the LR-bound, the anticommutator has a much richer behavior that can be compared to the bound we derived. First of all, it exhibits a clear light cone for both exponentially and algebraically decaying initial correlations, as expected.

In the space-like regime (large distance and small time), we find a correlation buildup which is algebraic for the zero temperature case with a power law determined by the initial state. This result does not saturate the bound derived in section 2 but has a slower growth. In the case of non-zero temperature, we dicussed the appearance of the thermal correlation length also outside of the light cone where the correlation buildup is exponential, yet again closely related to the initial state.

The situation *close to the light cone* is much more interesting. We find that the correlation func-

tion grows algebraically fast in that region independently on the initial state. The exponent of this growth is exactly the one that controls the spatial decay of the observable in the initial state at zero temperature. Comparing the exact calculation to the general bound we derived before, we find that the correlation buildup in the Luttinger liquid is very close to the maximal allowed buildup. Moreover, the connection between the initial state and the time evolution suggested by the bound is perfectly respected in the Luttinger liquid.

Finally, in the time-like region, the bound diverges exponentially fast, meaning we cannot use it to describe the real-time evolution. However, our exact calculations can be interpreted via the quasiparticles approach developed by Cardy and Calabrese. The spectrum of the excitations is linear and consequently they all have the same velocity. From our calculations we find that another ballistic signal with vanishing correlations is present inside the time-like regime. This is due to the fact that the light cone and the relaxation value (infinite-time value) of the anti-commutator have opposite signs. This confirms that even in a model with an extremely simple spectrum, observables can show a non-trivial behavior that cannot be guessed by more general results like the Lieb-Robinson bound.

## Acknowledgments

We are grateful for valuable discussions with M. Schmitt.

**Funding information**     The authors gratefully acknowledge financial support from the Deutsche Forschungsgemeinschaft (DFG) through SFB/CRC 1073 (Project B03).

## A   Optimization of the variables $l_A$ and $l_B$

We start our analysis from the simple case of equal time correlation function, where it is easy to grasp the ideas and then extend it to non-equal time. We want to find the parameter $l$ that minimizes the bound on the right side in Eq. (7):

$$|\langle O_A(t)O_B(t)\rangle_c| \leq 2c\left(|A|+|B|\right)\exp\left(-\frac{l-vt}{\xi}\right) + \frac{\tilde{c}}{\tilde{a}^\beta + (L-2l)^\beta}. \tag{38}$$

The parameters $|A|$, $|B|$, $c$ and $\tilde{c}$ are assumed to have a comparable order of magnitude. The parameter $\xi$ depends on the range of the interactions of the Hamiltonian that drives the time evolution. Since we want to study extremely short-range interactions, we can assume $\xi$ to be small, compared to the distances and times we want to observe. This allows us to rewrite the previous equation in a more simple way, where the exponential function is some sort of wall function going up at $l = vt + \xi$. This has the effect of reducing the space where we look for the optimum of our bound to the region $vt + \xi < l < L/2$. Hence, we can approximate the exponential to be zero in this range and so we need just to minimize the algebraic function which has its minimum at the boundary, so at $l = vt + \xi$.

We can then plug this result in the previous equation and obtain a simple answer for the optimized function:

$$|\langle O_A(t)O_B(t)\rangle_c| \leq \frac{\tilde{c}}{\tilde{a}^\beta + (L-2vt)^\beta}, \tag{39}$$

where we neglected $\xi$ because it is a negligible contribution.

This argument can be generalized easily to the non-equal time correlation function where now the minimization is given by two parameters $l_A$ and $l_B$. There, the same limit for the exponentials as before can be taken and it again acts as a wall function constraining the minimization

region to $l_A > vt + \xi$ and $l_b > vt_0 + \xi$. The minimum of the algebraic decay is then attained in the most far away point from the line $l_A + l_B = L$ which is, where its maximum is located so at $(l_A = vt + \xi, l_B = vt_0 + \xi)$. In this region the exponential is again vanishing due to the assumption of a very small $\xi$ parameter that results from the short interaction range. It is then possible to write the bound as:

$$|\langle O_A(t) O_B(t_0)\rangle| \leq \frac{\tilde{c}}{\tilde{a}^\beta + |L - v(t + t_0)|^\beta}. \tag{40}$$

This result is a posteriori confirmed by the exact calculation on the Luttinger Liquid model, which has extremely short-range interactions, namely point-like, which then confirms our predictions. Clearly, faster decays of the correlation function are possible, since the bound is an upper limit.

In conclusion, if we consider distances much larger than the average interaction range, the exponential decay is negligible and the optimal bound is determined just by the initial correlation function. The optimal bound is finally written in a compact form involving the initial correlation function, namely

$$|\langle O_A(t) O_B(t_0)\rangle| \leq \mathscr{C}^{\text{ini}}(L - v(t + t_0)). \tag{41}$$

# B  Generalization of the result by Brayvi *et al.* to non-equal times

It is possible to generalize the theorem which assumes an initial exponential decay of correlations found by Brayvi *et al.* in Ref. [36] to non-equal times. If the initial decay is given by

$$|\langle O_A(0)O_B(0)\rangle_c| \leq \tilde{c}\exp(-L/\chi), \tag{42}$$

one can perform a derivation analogous to Brayvi's and ours and find that

$$|\langle O_A(t)O_B(t_0)\rangle_c| \leq 2c|A|\exp\left(-\frac{l_A - vt}{\xi}\right) + 2c|B|\exp\left(-\frac{l_B - vt_0}{\xi}\right)$$
$$+ \tilde{c}\exp\left(-\frac{L - (l_A + l_B)}{\chi}\right). \tag{43}$$

With the optimal values $l_A = (\xi L + v((\xi + \chi)t - \xi t_0))/(2\xi + \chi)$ and $l_B = (\xi L + v((\xi + \chi)t_0 - \xi t))/(2\xi + \chi)$ one finds

$$|\langle O_A(t)O_B(t_0)\rangle_c| \leq \bar{c}(|A| + |B|)\exp\left(-\frac{L - v(t + t_0)}{2\xi + \chi}\right). \tag{44}$$

Setting $t_0 = t$ recovers the result by Brayvi *et al.*. The typical length of the leaks outside of the light cone depend on both the initial scale $\chi$ and the Lieb-Robinson scale $\xi$. The full derivation is analytically exact and is explained in the following.

The expression that we want to minimize is Eq. (43). In order to do that we take the gradient with respect to $l_A$ and $l_B$ obtaining the following two equations:

$$\partial_{l_A}|\langle O_A(t) O_B(t_0)\rangle_c| = -\frac{2c|A|}{\xi}\exp\left(-\frac{l_A - vt}{\xi}\right) + \frac{\tilde{c}}{\chi}\exp\left(-\frac{L - (l_A + l_B)}{\chi}\right) = 0, \tag{45}$$

$$\partial_{l_B}|\langle O_A(t) O_B(t_0)\rangle_c| = -\frac{2c|B|}{\xi}\exp\left(-\frac{l_B - vt_0}{\xi}\right) + \frac{\tilde{c}}{\chi}\exp\left(-\frac{L - (l_A + l_B)}{\chi}\right) = 0. \tag{46}$$

We have two equations and we want to use them to fix the two parameters $l_A$ and $l_B$ as functions of $L$ and $t$.

In order to minimize the gradient needs to vanish such that the difference between the two previous equations yields

$$|A| \exp\left(-\frac{l_A - vt}{\xi}\right) = |B| \exp\left(-\frac{l_B - vt_0}{\xi}\right). \tag{47}$$

This directly leads to

$$l_B - l_A = \xi \ln\left(\frac{|B|}{|A|}\right) + v(t_0 - t). \tag{48}$$

We can then use one of the previous equations to determine the missing parameters. Multiplying both sides of Eq. (45) by

$$\exp\left[\left(\frac{1}{\xi} + \frac{1}{\chi}\right)l_A - \frac{1}{\chi}l_B\right], \tag{49}$$

we obtain the following equation:

$$\exp\left[\left(\frac{2}{\chi} + \frac{1}{\xi}\right)l_A\right] = \frac{2c\chi|A|}{\tilde{c}\xi} \exp\left[\left(\frac{\xi L + \chi vt}{\chi \xi}\right) - \frac{l_B - l_A}{\chi}\right]. \tag{50}$$

Together with the previous relation $l_A$ is now given as a function of the other parameters:

$$\left(\frac{2}{\chi} + \frac{1}{\xi}\right)l_A = \ln\left(\frac{2c\chi|A|}{\tilde{c}\xi}\right) + \frac{\xi L + \chi vt}{\chi \xi} - \frac{1}{\chi}\left(\xi \ln\left(\frac{|B|}{|A|}\right) + v(t_0 - t)\right). \tag{51}$$

Hence,

$$l_A = \frac{\xi}{2\xi + \chi}\left[\chi \ln\left(\frac{2\chi|A|c}{\tilde{c}\xi}\right) + \xi \ln\left(\frac{|A|}{|B|}\right)\right] + \frac{L - vt_0 + \left(1 + \frac{\chi}{\xi}\right)vt}{2 + \frac{\chi}{\xi}}. \tag{52}$$

Once we got that, we can easily find the other parameter using the identity:

$$l_B = l_B - l_A + l_A = \frac{L - vt + \left(1 + \frac{\chi}{\xi}\right)vt_0}{2 + \frac{\chi}{\xi}} + \frac{\xi}{2\xi + \chi}\left[\chi \ln\left(\frac{2c\chi|B|}{\tilde{c}\xi}\right) + \xi \ln\left(\frac{|B|}{|A|}\right)\right]. \tag{53}$$

One can then get $l_B$ from $l_A$ by switching $t$ to $t_0$ and $|A|$ to $|B|$ simultaneously.

These solutions are plugged into the previous expression to find:

$$\frac{l_A - vt}{\xi} = \frac{1}{2\xi + \chi}\left[L - v(t + t_0) + \chi \ln\left(\frac{2c\chi|A|}{\tilde{c}\xi}\right) + \xi \ln\left(\frac{|A|}{|B|}\right)\right] = \frac{L - v(t + t_0)}{2\xi + \chi} + \mathscr{C}_1. \tag{54}$$

The same can be done for the other coordinate:

$$\frac{l_B - vt_0}{\xi} = \frac{1}{2\xi + \chi}\left[L - v(t + t_0) + \chi \ln\left(\frac{2c\chi|B|}{\tilde{c}\xi}\right) + \xi \ln\left(\frac{|B|}{|A|}\right)\right] = \frac{L - v(t + t_0)}{2\xi + \chi} + \mathscr{C}_2. \tag{55}$$

In the last step also the sum of $l_A$ and $l_B$ is calculated to be

$$l_A + l_B = \frac{2\xi\chi}{2\xi + \chi} \ln\left(\frac{2c\chi\sqrt{|A||B|}}{\tilde{c}\xi}\right) + \frac{2L + \frac{\chi}{\xi}v(t + t_0)}{2 + \frac{\chi}{\xi}}, \tag{56}$$

such that one finds eventually:

$$\frac{L - (l_A + l_B)}{\chi} = \frac{L - v(t + t_0)}{2\xi + \chi} + \mathscr{C}_3, \tag{57}$$

where $\mathscr{C}_3 = \frac{2\xi}{2\xi+\chi} \ln\left(\frac{2c\chi\sqrt{|A||B|}}{\tilde{c}\xi}\right)$.

Plugging this into the previous equation yields the tightest possible bound:

$$|\langle O_A(t) O_B(t_0)\rangle| \leq \left(2c|A|e^{-\mathscr{C}_1} + 2c|B|e^{-\mathscr{C}_2} + \tilde{c}e^{-\mathscr{C}_3}\right)\exp\left(-\frac{L - v(t+t_0)}{2\xi+\chi}\right), \qquad (58)$$

which identifies a light-cone traveling at speed $v$ with exponentially decaying leaks. Adjusting the constants brings this form back to the form in Eq. (44).

For small times $t$ and $t_0$, we can see that this bound, where the optimization has been performed analytically, is still growing linearly, since

$$|\langle O_A(t) O_B(t_0)\rangle| \leq \left(2c|A|e^{-\mathscr{C}_1} + 2c|B|e^{-\mathscr{C}_2} + \tilde{c}e^{-\mathscr{C}_3}\right)\exp\left(-\frac{L}{2\xi+\chi}\right)\left(1 + \frac{v(t+t_0)}{2\xi+\chi}\right). \quad (59)$$

## C    Additional calculations for the explicit bound of the correlation function

To explicitly bound the correlation function in the case of the Luttinger model one needs to find a suitable estimation for the different contributions in Eq. (16). The ansatz is to bound every term using the cascade

$$\frac{(x + v(t+t_0))^2 - a^2}{((x + v(t+t_0))^2 + a^2)^2} \leq \frac{(x + v(t-t_0))^2 - a^2}{((x + v(t-t_0))^2 + a^2)^2}$$
$$\leq \frac{(x - v(t-t_0)))^2 - a^2}{((x - v(t-t_0))^2 + a^2)^2} \leq \frac{(x - v(t+t_0)))^2 - a^2}{((x - v(t+t_0))^2 + a^2)^2},$$

for $x \geq v(t+t_0)$. However, this is not always fulfilled, since for $x = v(t+t_0)$, for example, the upper bound is negative, whereas the "smallest" term can be positive. One way to fix this is to add the small offset $\sqrt{3}a$ to the range of allowed $x$ values, i.e. $x \geq v(t+t_0) + \sqrt{3}a$. For each estimation one can then multiply both sides with both denominators and subtract the smaller term in the equation above. This yields, for example,

$$((\tilde{x} + \sqrt{3}a)^2 - a^2)((\tilde{x} + 2v(t+t_0) + \sqrt{3}a)^2 + a^2)^2$$
$$- ((\tilde{x} + \sqrt{3}a)^2 + a^2)^2((\tilde{x} + 2v(t+t_0) + \sqrt{3}a)^2 - a^2) \geq 0,$$

where we introduced $\tilde{x} := x - v(t+t_0) - \sqrt{3}a$ such that $\tilde{x} \in \mathbb{R}_0^+$. Straightforward algebra then shows that all terms are positive or zero if the bound is tight. Thereby, we have demonstrated that it is possible to bound all terms as in Eq. (25) in the specified region.

This shift by $\sqrt{3}a$ results from the comparison of the four different terms over the allowed $x$-range in the equation above. The upper bound is maximal and larger than the other contributions at $x = v(t+t_0) + \sqrt{3}a$.

## D    Correlation function at finite temperature

For $T > 0$ the density-density correlation function in a finite sytem of size $L$ is given by

$$\langle n(x,t)n(0,t_0)\rangle_{\text{th}} = \langle n(x,t)n(0,t_0)\rangle_{T=0} + \frac{2}{4\pi^2}\sum_{s=1}^{\infty}\Bigg[(\gamma\sqrt{1+\gamma^2} - \gamma^2 - 1)$$
$$\times \Big(V(x + v(t-t_0), a + \beta v_F s) + V(x - v(t-t_0), a + \beta v_F s)\Big)$$
$$-(\gamma\sqrt{1+\gamma^2} - \gamma^2)\Big(V(x + v(t+t_0), a + \beta v_F s) + V(x - v(t+t_0), a + \beta v_F s)\Big)\Bigg], \quad (60)$$

with

$$V(\tau, a) = \frac{2\pi^2}{L^2} \frac{1 - \cosh\left(\frac{2\pi}{L}a\right)\cos\left(\frac{2\pi}{L}\tau\right)}{\left(\cosh\left(\frac{2\pi}{L}a\right) - \cos\left(\frac{2\pi}{L}\tau\right)\right)^2} \tag{61}$$

(cf. Eq. (17)). However, the infinite sum cannot be executed in a closed form unless you consider the thermodynamic limit $L \to \infty$ as it is done in section 4.2.

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
