# Peer review of "Analysis of the buildup of spatiotemporal correlations and their bounds outside of the light cone"

_SciPost Physics, doi:SciPost Phys. 5, 052 (2018)_

## Round 1 · Referee Report · Anonymous (Referee 2) · 2017-8-14

Strengths

1 The physics behind the observed effects is explained nicely.
2 Calculations are detailed.
3 Luttinger liquid theory is relevant for and accessible through experiments with ultra-cold quantum gases,

Weaknesses

1 The general bounds in section 2 are not at all surprising. Given that LR-bounds are bounds on the operator norm of certain Heisenberg picture evolved observables it is immediately obvious that they imply that non-equal time
correlations can be bounded by in terms of functions equivalent to bounds on the correlations present in the initial state evaluated with at suitably shifted distances.
2 The general results should be at least presented as proper theorems so that they can be easily cited and reused.
3 The Level of detail is sometimes excessive and some of the lengthy calculations would be better presented in an appendix so that the main text could focus more on results and their discussion.
4 It is unfortunately quite noticeable that non of the authors is a native speaker.

Report

In the manuscript "Analysis of the buildup of spatiotemporal correlations and their bounds outside of the light cone" the authors use existing Lieb-Robinson bounds to derive bounds on the non-equal time correlation functions in terms of the correlations present in the initial state and then calculate such correlation functions exactly in a Luttinger liquid model. They find that the bounds outside the light cone of the LB-bound are essentially tight, and discuss the non-trivial behavior inside the light cone for their specific model, which is not constrained by the general bounds.

As I explained under "Weaknesses" above, for me the general results are not at all surprising. In fact, while the authors claim their results only for the case of initial power-law decaying correlations, I think that any bound on the correlations in the initial state should imply a bound with similar functional form (as long as the leakage outside the light cone is sub-dominant) on the non-equal time correlations. As SciPost Physics says of itself that it "publishes outstanding-quality research articles" and that it requires that "Articles provide in-depth, detailed reports of groundbreaking research within one or more subject areas" I think that this article is not above threshold for SciPost Physics. The nice and detailed calculations for the Luttinger liquid model do not change this assessment.

Should the manuscript be considered for publication despite these reservations of mine, then at least the points listed under "Requested changes" must be carefully addressed.

Requested changes

1 Page 1: "of this important parameter is anyway not fixed" remove the anyway 2 Page 1: "some of the requirements were loosened by several improvements and generalizations.[3, 4, 5, 6]" The citations should come before the full stop. This has happened in several other places not only on this page. 3 Page 1: "slower than the one predicted by the bound" the bound is just a bound it hence does not "predict" anything. It "allows" certain behavior, hence this should be changed to "slower than the one allowed by the bound" 4 Page 2: "These are directly con-nected to the anticommutator" How they are connected is only explained much later in Section 6. Please add a hint for the reader here. 5 Page 5: Between Eq. (7) and (8) it is said that l_A and l_B are optimized, but is the choice for l_A and l_B really known to be optimal? If so, could the authors please explain why this is the optimal choice? 6 Page 5: "need not to be constant" -> "need not be constant" 7 Page 6, last paragraph: Here it is claimed that a result of by Brayvi et al. can be extended to non-equal time correlation functions. Given that also everywhere else the paper manuscript is not shy of presenting lengthy calculations, the proof of claimed result should be presented, at least in an appendix, instead of just claiming that "In this case, one can show that ...". 8 Page 8: "It is given as" -> "It is given by" 10 Page 10: The fourth line starts with an inverted question mark, which doesn't exist in the English language and does not make sense here. 11 Page 10: "or both light cones (see also Ref. [[48]])." Here, as well as in several other places citations are typeset with double brackets [[...]] instead of single brackets [...]. This should be fixed. 12 Page 11: Eq. (22) and (23) overlap into the page margin. 13 Page 12: The equation without number between (25) and (26) has duplicate <= signs at the beginning and end of each line and uses / instead of \frac to typset fractions, which makes it hard to read, this should be fixed. More importantly: I do see that both the enumerator and the denominator are individually larger in the first line compared to the last line and I also see that the denominator decreases quadratically, while the enumerator decreases only linearly. Still, this only ensures that the fraction increases if the denominator is larger than 1, no (only then the slope of a parabola is steeper than that of a linear line)? Is this always the case and if so why and where are the necessary assumptions stated? 14 Page 12: Eq. (26) overlaps into the margin. 15 Page 13: "The average is taken with respect to the ground state" better say "expectation value" instead of average. Probably it would be good to give the expectation value in Eq. (27) a subscript such as "therm,kin" or something like that. 16 Page 13: Equation (28) must be changed. While it is pretty clear what the authors mean, one simply can not define a function to be equal to itself minus another function. This is completely nonsensical. 17 Page 14: Since Eq. (31) is only approximately true it shouldn't be stated as an equality, but a proper approximate sign should be used. 18 Page 14: "if you focus on the de" This is inappropriately colloquial language 19 Figures 2, 3 and 4: Panels (b) should have a label on their y axis 20 Figure 3: What is the blue dashed line? I couldn't find an explanation anywhere in the caption. 21 Page 19: "whereas the LR-bound is an operator norm and thus" -> "whereas the LR-bound is an upper bound on the operator norm and thus" (otherwise the statement is completely nonsensical. 22 Page 20: There is an Appendix A that consists of just a single formula without any text and I am not even sure it is referenced from anywhere in the main text. This can obviously not stay like that.

  • validity: good
  • significance: ok
  • originality: ok
  • clarity: good
  • formatting: below threshold
  • grammar: below threshold

---

## Round 1 · Referee Report · Anonymous (Referee 1) · 2017-8-14

Strengths

The paper treats a timely and interesting physical questions. It gives very general bounds together with the discussion of one specific example.

Weaknesses

See requested changes

Report

In the manuscript 'Analysis of the buildup of spatiotemporal correlations and their bounds outside of the light cone' the authors derive a bound for the correlations functions of two local, but spatially separated observables at different times using the Lieb-Robinson bound. They further discuss how this bound is taken in the exaclty solvable Luttinger model after a sudden quench from the non-interacting to the interacting model.
The manuscript is well written and addresses an interesting question in a solid way. Therefore, I would recommened it for publication after the requested questions/points are addressed.

Requested changes

1) In order to derive equation (8) the authors use an ansatz for l_A and l_B. I do not understand why they can make this ansatz and what it implies. A few more lines of explanation would be helpful.

2) A comparison of the bound with previous results in the conclusion would be helpful. For example the Ref. 56 (never mentioned in the text) and reference 21 show closely related results.

3) Ref. 23 should be substituted by Ref. 56 (at least I do not understand the relation of Ref. 23 to the current manuscript).

Concerning the layout: citations are always inserted after the full-stop, sometimes they have double brackets, spell problems (capital letters) in the titles of the citations should be resolved, the equations are not always nicely set typos: 'his newly defined' -> this newly defined inversed ? occurs infront of 'From its functional form' I have not found refernces 51-56 in the text.

  • validity: high
  • significance: good
  • originality: good
  • clarity: high
  • formatting: acceptable
  • grammar: good

Author:  Lorenzo Cevolani  on 2018-08-21  [id 307]

(in reply to Report 1 on 2017-08-14)

Response to referee 1: 1) We specified what assumptions are made such that the bound is optimal and elaborate the derivation appendix A. 2) We added a comparison of the bound in chapter 2. 3) Changed the reference to Ref. 56. Removed Ref. 23.. - Fixed the citations and their titles. Fixed the equations. - Fixed typos - Removed the references 51-56 from the bibliography

Response to referee 2: 1) Page 1: Removed the "anyway". 2) Page 1: Changed all citations to look properly. 3) Page 1: Changed to "allowed". 4) Page 2: Added a hint. 5) Page 5: We provided more details about the choice of the optima l_A and l_B parameters in the Appendix A. The main point is that in some limits the exponentially decaying function of the Lieb-Robinson bound can be ignored after a constraining the optimization space of the previous variables. In this way it is possible to find an explicit solution of the minimisation problem. 6) Page 5: Corrected. 7) Page 6: Extended the proof and moved it to the appendix. 8) Page 8: Corrected. 10) Page 10: Corrected. 11) Page 10: All wrong references have been fixed. 12) Page 11: All equations that overlapped into the margin have been rearranged. 13) Page 12: Fixed the duplicate symbols. The estimation has been made more precise and moved to the appendix. All necessary assumptions have been added to the text. 14) Page 12: Fixed. 15) Page 13: Changed to "expectation value". 16) Page 13: Removed the misleading equation and added an explanation in the text. 17) Page 14: The equation describes the function that the approximation in the deep space-like region converges to. In that sense, it can be rather understood as a definition of the space-like behavior In the deep space-like regime. 18) Page 14: Fixed. 19) Figures 2,3,4: Added y-axis label. 20) Figure 3: Added description of blue dashed line. 21) Page 19: Fixed. 22) Page 20: Added description of the formula.

---

## Round 3 · Referee Report · Anonymous (Referee 3) · 2018-11-5

Strengths

1 - Clear message, clear presentation
2 - technically sound

Weaknesses

1 - The research presented in this work is not highly original, but rather incremental beyond existing work.

Report

The paper closes an open gap in the existing Lieb-Robinson (LR) literature, as it proves LR bounds for unequal times also in the case of initial states with power-law correlations. The paper is technically sound, clearly presented and can be published in SciPost Physics. I recommend that the authors have a look at the recent experimental work https://journals.aps.org/prx/abstract/10.1103/PhysRevX.8.021070, which illustrates
LR bounds in two-dimensional systems, and which seems to have escaped the authors attention.

Requested changes

none

---

## Round 3 · Referee Report · Anonymous (Referee 4) · 2018-11-8

Strengths

N/A

Weaknesses

N/A

Report

This report is entered by the editor for one of the original referees that submitted a report on the resubmission by email, and not via the online interface. They are happy with the changes made, and recommend acceptance.

Requested changes

N/A

---

## Round 3 · Author Response

Response to referee 1:

1) We specified what assumptions are made such that the bound is optimal and elaborate the derivation appendix A. 2) We added a comparison of the bound in chapter 2. 3) Changed the reference to Ref. 56. Removed Ref. 23.. - Fixed the citations and their titles. Fixed the equations. - Fixed typos - Removed the references 51-56 from the bibliography

Response to referee 2: 1) Page 1: Removed the "anyway". 2) Page 1: Changed all citations to look properly. 3) Page 1: Changed to "allowed". 4) Page 2: Added a hint. 5) Page 5: We provided more details about the choice of the optima l_A and l_B parameters in the Appendix A. The main point is that in some limits the exponentially decaying function of the Lieb-Robinson bound can be ignored after a constraining the optimization space of the previous variables. In this way it is possible to find an explicit solution of the minimisation problem. 6) Page 5: Corrected. 7) Page 6: Extended the proof and moved it to the appendix. 8) Page 8: Corrected. 10) Page 10: Corrected. 11) Page 10: All wrong references have been fixed. 12) Page 11: All equations that overlapped into the margin have been rearranged. 13) Page 12: Fixed the duplicate symbols. The estimation has been made more precise and moved to the appendix. All necessary assumptions have been added to the text. 14) Page 12: Fixed. 15) Page 13: Changed to "expectation value". 16) Page 13: Removed the misleading equation and added an explanation in the text. 17) Page 14: The equation describes the function that the approximation in the deep space-like region converges to. In that sense, it can be rather understood as a definition of the space-like behavior In the deep space-like regime. 18) Page 14: Fixed. 19) Figures 2,3,4: Added y-axis label. 20) Figure 3: Added description of blue dashed line. 21) Page 19: Fixed. 22) Page 20: Added description of the formula.

---

## Round 3 · List of Changes

1) In Appendix A we present in more detail the derivation of our bound for non-equal time correlation functions.
2) In Appendix B we extend the result of Bravy et al (Phys. Rev. Lett. 97, 050401, 2006) to non equal time correlation functions.
3) We fixed typos in the manuscript
4) We fixed the bibliography

---

## Editorial Decision

published